# Lumacaftor and Matrine: Possible Therapeutic Combination to Counteract the Inflammatory Process in Cystic Fibrosis

**DOI:** 10.3390/biom11030422

**Published:** 2021-03-13

**Authors:** Michela Pecoraro, Silvia Franceschelli, Maria Pascale

**Affiliations:** Department of Pharmacy, University of Salerno, Via Giovanni Paolo II 132, 84084 Fisciano, Salerno, Italy; mipecoraro@unisa.it (M.P.); pascale@unisa.it (M.P.)

**Keywords:** Cystic fibrosis (CF), oxidative stress, inflammation, CFTR rescue, Vx-809 (Lumacaftor), Matrine

## Abstract

Cystic fibrosis is a monogenic, autosomal, recessive disease characterized by an alteration of chloride transport caused by mutations in the CFTR (Cystic Fibrosis Transmembrane Conductance Regulator) gene. The loss of Phe residue in position 508 (ΔF508-CFTR) causes an incorrect folding of the protein causing its degradation and electrolyte imbalance. CF patients are extremely predisposed to the development of a chronic inflammatory process of the bronchopulmonary system. When the cells of a tissue are damaged, the immune cells are activated and trigger the production of free radicals, provoking an inflammatory process. In addition to routine therapies, today drugs called correctors are available for mutations such as ΔF508-CFTR as well as for others less frequent ones. These active molecules are supposed to facilitate the maturation of the mutant CFTR protein, allowing it to reach the apical membrane of the epithelial cell. Matrine induces ΔF508-CFTR release from the endoplasmic reticulum to cell cytosol and its localization on the cell membrane. We now have evidence that Matrine and Lumacaftor not only restore the transport of mutant CFTR protein, but probably also counteract the inflammatory process by improving the course of the disease.

## 1. Introduction

Cystic fibrosis (CF) is an autosomal recessive disease involving mucus and sweat-producing cells, that though it affects multiple organs, the lungs are the most severely compromised leading to death in 90% of patients [1]. CF has an incidence of 1 in 2500 live births with a predominance in those of Northern European descent [2]. The disease is due to mutations in the gene encoding the cystic fibrosis transmembrane conductance regulator (CFTR) protein, which is expressed in a large number of epithelial and blood cells [3]. The most common genetic mutation is in Phe508Δ gene [4], caused by a deletion of three base pairs (c.1521_1523delCTT; p. Phe508Δ), with the consequent loss of a phenylalanine. It has been estimated that 68% of cases of CF patients have this deletion and search for other alleles is in progress [5].

CFTR protein is transported co-translationally into the lumen of the endoplasmic reticulum (ER), where it is N-glycosylated and folded. Upon folding, CFTR acquires a transport-competent conformation allowing its delivery to the Golgi apparatus where it is introduced into the plasma membrane, acting as a channel protein [6]. Although CFTR functions mainly as a chloride channel, it has been shown to have several other regulatory roles, including regulation of the outwardly rectifying chloride channel, ATP channels, intracellular vesicle transport, acidification of intracellular organelles, and inhibition of endogenous calcium-activated chloride channels [7].

Studies of the turnover of wild type and ∆F508 mutant CFTR have shown that up to 75% of the newly synthesized wild type CFTR is degraded, while ∆F508 variant protein is almost completely degraded [8]. ∆F508 mutant protein remains for a longer time than the wild type in the ER where Hsc-70, an important molecular chaperon, facilitates CFTR ubiquitination and increases retrograde transport back to the cytosol where eventually it is degraded by the proteasome [9,10]. It has been shown that ∆F508 mutation impairs folding of the nucleotide-binding domain in vitro [11] and that the portion of ∆F508 CFTR that folds properly displays ion channel activity indistinguishable from wild type [12]. Overall, these data are consistent with the notion that the impaired folding of the protein plays a fundamental role in the molecular pathology of the disease. Concerning the respiratory system, airway obstruction, resulting from the thick mucus and reduced clearance of inhaled particles, promotes persistent infection and chronic inflammation that are the major causes of death. However, the origin of inflammation in CF has been a matter of debate and, recent data suggest that the retention of the misfolded protein could play a key role in the development and maintenance of the inflammatory response [13]. Clearly, the host response, essential to contain the infection, also translates into an oxidative and proteolytic environment which at the same time contributes to perpetuating the proinflammatory condition of the respiratory tract [14]. Oxidative stress determines a deleterious effect of cell function in pathologies associated with inflammation [15]. Further, in agreement with several studies [16,17], defects of CFTR, as a function of its localization, also contribute to the endogenous activation of NF-κB, and consequently to an excessive production of the proinflammatory cytokines. Velsor and co-workers have reported the presence of high intracellular levels of hydrogen peroxide, reflecting oxidative stress, which are also associated with a CFTR-deficient state [18]. Moreover, an excessive production of proinflammatory cytokines increases reactive oxygen species (ROS) production, perpetuating the vicious circle of inflammation in CF [19], enhancing the susceptibility to apoptosis [20].

In the last decades, most of CF therapies were directed against lung infection and inflammation. Nevertheless, novel approaches directly targeting the basic CFTR default with mutation-specific therapies, have recently emerged [21]. Research has mainly focused on the identification and development of modulators capable of solving gating and trafficking problems, by increasing the amount of CFTR delivered to the cell surface [22,23,24,25].

Recently, the Food and Drug Administration (FDA) approved a new generation of CFTR modulator: TRIKAFTA™ (Vertex Pharmaceuticals), the combination of Vx-661 (tezacaftor) plus Vx-445 (elexacaftor) corrector, that together cooperate in processing misfolded CFTR protein to the cell membrane and Vx-770 (ivacaftor) potentiator which increases channel opening, improving synergistically the treatment of patients with at least 1 ΔF508 allele [26].

Studies in vitro show that Trikafta TM significantly improves ΔF508-CFTR protein trafficking, processing and chloride transport compared to any two of these agents in dual combination. In CF patients, this combination had an acceptable safety and side effect profile with mild or moderate adverse effects [27,28].

The “corrector” Lumacaftor (Vx-809) acts on trafficking, promoting greater CFTR localization on the membrane cellular, while Matrine, a quinolizidine alkaloid [29,30], modulates the molecular chaperone activity in the cell, improving maturation of the mutant CFTR protein [31,32]. Matrine, which is extracted from a Chinese traditional herb *Sophora flavescens*, has exhibited various pharmacological effects such as against viral hepatitis infection, antifibrosis, and anti-inflammation, and it has also been suggested that Matrine exerts anti-oxidative effects [30]. Guo and co-workers showed that Matrine is a novel inhibitor of the TMEM16A, a molecular component of calcium-activated chloride channels (CaCCs) [33].

Here, we show that the synergic effect of CFTR corrector Vx-809 in combination with the Matrine, is necessary to allow a significant rescue of CFTR trafficking and gating.

Previous studies conducted by Stanton have already shown that treatment with Vx-809 alone has no significant effect on the constitutive secretion of IL-6 and IL-8 [34].

The combined use of these molecules represents an important basis for new therapeutic opportunities to reduce oxidative stress, possibly with a consequent reduction of the inflammatory process and amelioration of CF disease.

## 2. Materials and Methods

### 2.1. Reagents

Vx-809 (S1565) was purchased from Selleckchem (Houston, TX, USA). The anti-Hsc70 (sc-7298), anti-Hsp90 (sc-7947), anti-MnSODIII (sc-271170), anti-tubulin (sc-32293), anti-iKKα (sc-7606), anti-IkBα (sc-203), anti-caspase 4 (sc-1229), and Na^+^/K^+^ ATPase (sc-48345) monoclonal antibodies were obtained from Santa Cruz Biotechnology Inc. Mouse monoclonal anti-CFTR antibodies were purchased from Millipore (M3A7) or Abcam (ab2784). Secondary antibodies (anti-rabbit, A120-101P and anti-mouse, A90-137P) were purchased from Bethyl Laboratories (Montgomery, TX, USA). 

Texas red-conjugated secondary antibody (T6390 and PA1-28662) was bought by Thermo Fisher Scientific (Waltham, MA, USA). Matrine, H_2_DCF-DA and propidium iodide were purchased from Sigma-Aldrich (St. Louis, MO, USA).

### 2.2. Cell Culture

Adenocarcinomic human alveolar basal epithelial cells (A549) stably overexpressing wild type (wt) CFTR or ΔF508-CFTR were kindly provided by Dr Luis J. V. Galietta (Istituto Giannina Gaslini and Centro di Biotecnologie Avanzate, Genova, Italy). Cells were grown to confluence in Dulbecco’s modified Eagle’s Medium (DMEM)-Ham’s F-12 (1:1) supplemented with 10% fetal bovine serum (FBS), 2 mM L-glutamine, and antibiotics (25 U/mL penicillin and 25 U/mL streptomycin) under an atmosphere of 95% air/5% CO_2_ at 37 °C.

### 2.3. Experimental Protocol

Human lung cells were treated with Vx-809 (2 µM) and Matrine (300 µM) alone and exposed to combined treatment with Vx-809 (2 µM) and Matrine (300 µM) for 48 h in DMEM)-Ham’s F-12 (1:1) 10% FBS.

### 2.4. Indirect Immunofluorescence Analysis 

For the immunofluorescence assay, cells (2 × 10^4^ per well) were seeded on cover slips in 12 well plates and allow to grow for 24 h; thereafter, cells were treated as described above. Then, cells were fixed with 3.7% formaldehyde in PBS for 15 min. After blocking with BSA and PBS for 1 h, cells were incubated with mouse anti-CFTR antibody (ab2784) O/N at 4 °C. 

Cells were then washed three times with PBS and Texas red-conjugated secondary antibody was added for 1 h, and DAPI was used for counterstaining of nuclei. Coverslips were finally mounted in a mounting medium and fluorescent images were taken under a Confocal Laser Scanning Microscope (Leica TCS SP8, Heidelberg, Germany) [35].

### 2.5. Cytosol and Membrane Extracts

Briefly, cells (1 × 10^6^) were washed twice with PBS, detached with scraper in PBS, and centrifuged for 5 min at 2000 rpm at 4 °C. Afterwards, cells were lysed in 4 mL of buffer A (Tris HCl 50 mM, pH 7, 4; sucrose 1 M; DTT 1 M; protease inhibitors, EDTA 100 mM), sonicated (5 s pulse–9 s pause for 2 min, amplitude 52%). The samples were ultra-centrifuged for 1 h at 32,500 rpm at 4 °C, and new obtained supernatants represent cytosol extracts. Each resultant pellet was dissolved in 4 mL of buffer A and ultra-centrifuged for 1 h at 32,500 rpm at 4 °C. After, the final pellets were resuspended in 250 μL of buffer B (Tris HCl 50 mM, pH 7, 4; DTT 1 M; EDTA 100 mM; Triton X-100 1% and sucrose 1 M) and left on orbital shaker at 4 °C overnight. Finally, the solution was centrifuged for 30 min at 13,000 rpm at 4 °C: the supernatants represent membrane extracts. Na^+^/K^+^ ATPase was used as marker for the membrane fraction [36].

### 2.6. Total Protein Extraction and Western Blot Analysis 

Cells were seeded (3.0 × 10^4^ cells) in 12-well tissue culture plates and were grown for 24 h before use. Then, the medium was replaced with fresh medium and cells were treated as described in experimental protocol.

Total proteins were extracted from cells by freeze/thawing in lysis buffer (containing 150 mM NaCl, K^+^ Hepes pH 7.5 20 mM, 1 mm EDTA, IGEPAL 1% and protease inhibitor cocktail). Thereafter, the cells were centrifuged at 14,000 rpm for 15 min at 4 °C. Protein concentration was determined by a Bradford assay and 10 µg of total protein were run on an 8–10% acrylamide gel and separated by SDS-PAGE and transferred to nitrocellulose membranes using a minigel apparatus (Bio-Rad Laboratories, Richmond, BC, Canada). Blots were then blocked in Tris-buffered saline, containing 5% nonfat dry milk for 1 h at room temperature and incubated overnight with specific primary antibodies at 4°C with slight agitation. Tubulin was used as loading control. After washes in PBS/0.1% Tween, the appropriate secondary antibody was added for 1 h at room temperature. Antigen–antibody complexes were then visualized using an enhanced chemiluminescence (ECL) immunoassay in LAS 4000 (GE Healthcare, Chicago, IL, USA). Images were quantified by ImageJ Software.

### 2.7. Measurement of Intracellular Reactive Oxygen Species (ROS) 

ROS formation was evaluated by the probe 2′,7′-dichlorofluorescin diacetate (H_2_DCF-DA). H_2_DCF-DA diluted to serum-free DMEM-Ham’s F-12 to a final concentration of 10 μM. The complete medium was removed, and the cells were washed twice with phosphate-buffered saline (PBS). The diluted H_2_DCF-DA medium was added to the cells and incubated at 37 °C for 30 min. After incubation, cells were washed twice with PBS, then collected and resuspended in PBS. The fluorescence value was measured by flow cytometry and analyzed with Cell Quest software (BD Bioscience, San Jose, CA, USA). The parameter settings included an excitation wavelength of 488 nm and an emission wavelength of 525 nm.

### 2.8. Flow Cytometry Analysis

Cytosolic IkBα, cytosolic CFTR, or caspase 4 were checked by fluorescence-activated cell sorting (FACSscan; BD Bioscience, San Jose, CA, USA). A549 cells were cultured in a 12-well plate (2.5 × 10^3^ cells/well) and allowed to grow for 24 h; thereafter, the medium was replaced with fresh medium and cells were treated as described. After incubation period, cells were harvested and treated with fixing buffer (containing 2% FBS and PBS in the presence of sodium azide 0.1%, 4% formaldehyde) for 20 min and then permeabilized with a specific buffer for 30 min (containing 2% FBS and PBS in the presence of sodium azide 0.1%, 4% formaldehyde and Triton X-0.1%). Specific primary antibodies were added and incubated for 60 min. Anti-mouse or anti-goat Texas-Red were used as a secondary antibody. Cells were then washed twice with fixing buffer and then analyzed by FACS. Results obtained were analyzed by means of Cell Quest software. Data are shown as percentage of positive cells.

### 2.9. Determination of Hypodiploid Nuclei

Hypodiploid nuclei was analyzed using propidium iodide (PI) staining by flow cytometry [37]. Briefly, cells were cultured (2.5 × 10^3^ cells/well) in a 12-well plate and allowed to grow for 24 h and treated as previously described. After the treatment period, cells were washed twice with phosphate buffered saline (PBS) and incubated in 500 µL of a solution containing 0.1% Triton X-100, 0.1% sodium citrate, and 50 µg/mL PI, at 4 °C for 30 min in the dark. The PI-stained cells were analyzed by means of FACscan using Cell Quest software. Cell debris were excluded from the analysis by raising the forward scatter threshold, and DNA content of the nuclei was registered on a logarithmic scale. Results are expressed as percentage of hypodiploid region.

### 2.10. Statistical Analysis

Statistical analysis was performed with GraphPad Prism7 software (GraphPad Software Inc., San Diego, CA, USA). Data are represented as mean ± S.E.M. of at least three independent experiments, each performed in duplicate. Statistical analysis was carried out using nonparametric Mann–Whitney U test. *p* values from 0.01 to 0.05 were considered as statistically significant.

## 3. Results

### 3.1. Vx-809 and Matrine Co-Treatment Increases CFTR Localization in Membrane

A549 cells expressing ∆F508-CFTR (hereinafter referred to as A549∆) cotransfected with halide-sensitive yellow fluorescent protein (YFP)-H148Q/I152L/F46L [38].

From the images obtained by confocal microscopy (Figure 1A), it is possible to observe that in A549∆ the single treatment with Matrine (300 µM) or with Vx-809 (2 µM) induces a change in the localization of the CFTR protein (colored red) from a diffuse localization in an area that was identified as ER-Golgi.

Finally, immunofluorescence analysis revealed indirectly that, combined treatment of Matrine and Vx-809, achieved significant reduction of yellow fluorescence in favor of red fluorescence in conjunction with the change in cellular localization of CFTR (Figure 1A).

Intracellular staining, by using flow cytometry analysis in A549∆, showed a significant (*p* < 0.05) increase in cytosolic CFTR levels in co-treated cells (Figure 1B). 

Western blot analysis on A549∆ membrane extracts revealed that Vx-809 and Matrine administration induces an increase CFTR C band. These data would confirm the CFTR presence in its glycosylated form and its localization in the membrane (Figure 1C).

### 3.2. Vx-809 and Matrine Co-Treatment Interferes in the Molecular Trafficking

Hsc70 is a pivotal molecular chaperon necessary for the ubiquitin-directed proteasome-mediated degradation of several cellular proteins, such as CFTR [39]. Again, Matrine modulates the levels Hsc70 and allows ΔF508-CFTR to get to the membrane [31]. Accordingly, A549∆ were treated or not with Matrine and Vx-809. In the presence of Vx-809 and Matrine, a slight decrease in the Hsc70 expression versus untreated cells was observed, most likely due to an enhancement of the recovery action of mutant CFTR (Figure 2A).

Furthermore, Hsp90 interacts with CFTR, though it is not yet known precisely how it affects CFTR folding. Our results establish that Vx-809 and Matrine co-treatment causes a downregulation of Hsp90 levels in A549∆, suggesting that the co-treatment stabilizes the protein, without requiring additional interactions (Figure 2B).

### 3.3. Vx-809 and Matrine Co-Treatment Reduces CF-Induced Oxidative Stress

Overproduction of ROS leads to pathological effects caused by deleterious oxidative changes in cellular lipids, proteins, and DNA, as well as the formation of inflammatory proteins. This pathway is also strongly implicated in CF [40]. High levels of ROS have already been observed in untreated A549∆, indicating that oxidative stress is already present in this cell line. Flow cytometric analysis using DCHF-DA fluorescent probe showed that the co-treatment with Vx-809 and Matrine significantly reduced (*p* < 0.005) cytosolic ROS production (Figure 3A). As depicted in Figure 3B, WT-CFTR-expressing A549 cells (A549wt) showed low level of cytosolic ROS at all experimental points, confirming that mutant CFTR expression induces an increase in oxidative stress. Treatment with Matrine alone determines an increase in the production of cytosolic ROS, in accordance with various articles [41,42].

### 3.4. The Synergistic Effect of Vx-809 and Matrine Counteract ROS Signaling

Once the increase in intracellular ROS has been confirmed in this in vitro model, the various antioxidant systems activated in response to Vx-809 and Matrine have been investigated.

SODIII, an antioxidant enzyme, which cooperates in the defense mechanisms against free radicals, has been evaluated. Indeed, this enzyme is highly expressed in lungs and proinflammatory cytokines increase its expression, in culture and in animal models of lung injury [43]. In untreated A549∆ we observed high levels of SODIII necessary to neutralize ROS effect. Vx-809 and Matrine administration, as shown in Figure 3A, reduces ROS levels, this consequently entails a reduction of SODIII expression, confirming its reduction in oxidative stress conditions, as showed in Figure 4A.

It is known that the proinflammatory pathway is regulated by the nuclear factor-kB (NF-kB)/inhibitor of NF-kB (IkB), and that its activation can be induced by increased ROS levels [44]. 

Phosphorylation of IκBα is mediated by IKK [45], so we chose to analyze the expression of IKKα in cells treated as above.

As showed in Figure 4B, co-administration of Vx-809 and Matrine significantly reduces IKKα expression (*P* < 0.05) and increases IκBα levels as detected by flow cytometry. These data show that Vx-809 and Matrine co-treatment is effective in interfering with the NF-kB mediated pathway.

### 3.5. The “Corrector” of CFTR and the Quinolizidine Alkaloid Interfere with the Cell Death

Several lines of evidence convincingly show that lung epithelial cell death/apoptosis is one of the critical pathophysiologic events limiting normal lung repair and thereby facilitating pulmonary fibrosis [46]. It has also been proven that apoptosis can be induced by an increase of ROS levels and poorly folded proteins [47]. The image 5A show that, in the A549∆, the untreated samples already have high percentage of apoptosis due to high levels of incorrectly folded CFTR protein; our data demonstrated that, in both A549∆ and A549wt, Vx-809 and Matrine co-admnistration causes a significantly high reduction of apoptotic response (*p* < 0.001) (Figure 5A,B).

The expression of mutated proteins induces a stress on the ER [48] which very often results in the activation of the apoptotic process. Though several mechanisms have been reported as activating apoptotic signaling pathways, those that lead to ER stress-induced cell death in humans remain poorly circumstantiated.

Caspase 4 is localized in the ER membrane and is cleaved when there is an ER stress [49]. In fact, high levels of caspase 4 are present in the A549∆, even in the untreated cells only, in agreement with the activation of the reticular apoptosis canonical way. Flow cytometry analysis showed a significant reduction of caspase 4 levels in Vx-809 and Matrine co-treated cells (Figure 5C), assuming an improvement in CFTR trafficking to the membrane.

## 4. Discussion

CF is a common autosomal recessive disorder due to mutations of the CF transmembrane conductance regulator (CFTR) gene. Defects in CFTR lead to a lung disease that is characterized by airway obstruction, persistent bacterial infection, and vigorous neutrophilic inflammation. Over time, inflammatory responses escape homeostatic control, become excessive, and cause damage to host tissues [14].

One of the major pathogens present in CF patients is *P. aeruginosa*. Prolonged infections of this pathogen have been linked to chronic inflammation of the CF lung, aggravating damage to lung tissue, and leading eventually to respiratory failure.

The FDA has approved the Lumacaftor–Ivacaftor (ORKAMBI^®^) combination for patients with the ∆F508 CFTR mutation. However, clinical studies on ORKAMBI^®^ have shown that the response to treatments is variable between patients and the beneficial effects on lung function remain below expectation. However, exposure to *P. aeruginosa* has been shown to reduce Vx-809 mediated rescue of CFTR in human bronchial epithelial cells by increasing the expression of proinflammatory cytokines [34,50,51].

Several evidences showed that under pathophysiological conditions, and, also, in CF, activated neutrophils and epithelial cells release highly reactive molecules towards the extracellular space, such as ROS to attack and eliminate invasive pathogens [52]. 

Our experiments confirm the presence of ROS in cells overexpressing ΔF508-CFTR, which mimic the stress condition in CF.

In the present study, we evaluated whether the action of the synergistic treatment of Vx-809 and Matrine could restore the folding and trafficking of the CFTR protein, acting on the molecular chaperones involved, such as Hsc70 and Hsp90 and contrasting the main mechanisms involved in oxidative stress to hinder the establishment of the inflammatory process, closely related to the NF-κB and apoptosis pathway.

Hsc70/Hsp90 cytosolic chaperone systems contribute to the conformational and functional maintenance of the ΔF508-CFTR, resident in the plasma membrane as they facilitate nascent chains folding and the folding of denatured and aggregation-prone polypeptides, hiding exposed hydrophobic surfaces.

Hsp70 family members preferentially recognize the unfolded proteins, while Hsp90 chaperones bind to partially folded intermediates and represent the maintenance of the active conformation of their clients. It has been proposed that the buffering capacity of chaperones could improve genetic diversity [53].

Our data demonstrate that the co-treatment with Vx-809 and Matrine reduces Hsc70 and Hsp90 expressions, assuming an enhancement of the recovery action of mutant CFTR. 

Cytofluorimetric analysis showed in CFTR content in the cytosol of co-treated cells and immunofluorescence analysis revealed a change in cellular localization of CFTR. 

Expression of ∆F508 CFTR in the different experimental conditions was also confirmed by Western blotting analysis. The co-treatment with Vx-809 and Matrine reduces the ROS production in our experimental in vitro model. The decrease in ROS levels also translates into a reduction of enzymes involved in decreasing superoxide anion levels that damage cells with an excessive concentration, in particular of superoxide dismutases III (SODIII). Alterations in SODIII expression have been described in lung diseases, and it is associated with decreased neutrophil recruitment, suggesting an important role in regulating pulmonary inflammation.

Although no direct evidence has shown the involvement of SODIII in CF, the finding that it is highly expressed in airways raises the possibility that it may play a role in this pathology [43]. The inflammation triggered by oxidative stress is the cause of many chronic diseases. Additionally, the accumulation of defective CFTR proteins in the ER contributes to endogenous activation of NF-kB signaling pathway, which enter the nucleus and initiate transcriptional activities associated with inflammation [12,54]. Our results clearly show that the synergistic treatment of Vx-809 and Matrine induces a significant reduction in IKKα with a simultaneous increase in IkBα, thus indicating an inactivation of NF-κB pathway in our experimental model.

Previous studies have shown that NF-kB pathway controls apoptosis and inflammation in CF cells [3]. Indeed, mutant CFTR is recognized as a misfolded protein, and is trapped in the ER, causing stress. The excessive or long-termed ER stress results in apoptotic cell death, involving nuclear fragmentation and chromatin condensation [48].

It has been reported that the activation of caspase 4 in cultured human cells is precipitated by ER stress [54]. Moreover, caspase 4 may function mainly via the NF-κB signal pathway in inflammatory responses [55].

In this study, we show that Vx-809 and Matrine co-administration, acting on the folding and trafficking of the ∆F508 CFTR protein restore, reduces the apoptotic process, as shown by FACs analysis. Furthermore, countering the ER stress, cytofluorimetric analysis showed a reduction of caspase 4 levels, in our experimental model.

We believe that the synergistic treatment of Vx-809 and Matrine counteract the triggering of the inflammatory process (Scheme 1).

Although CFTR undoubtedly plays a salient role in the regulation of anions, and thus of fluid secretion through the mucosa of the airways, it is likely that other additional hydration mechanisms (other ion channels) exist even in the absence of normal CFTR function, as the clearance of mucus is altered but not completely inhibited. In CF lungs [56], particular interest is addressed to TMEM 16A, an important calcium activated chloride channel (CaCC), present in the epithelia of the airways whose physiological role; however, is not yet clear. Indeed, there are controversial literature data regarding its ability to improve anion secretion in CF. Thus, an unsolved question that remains to be answered is can activation or inhibition be a therapeutic approach for CF? 

More recently, data published by Guo and co-authors highlighted that Matrine inhibits TMEM16A [33], supporting the hypothesis that Matrine may also interfere with membrane ion channels as well as affect the intracellular trafficking of the mutant CFTR protein and improving its arrival at the membrane. This intriguing hypothesis opens up new and useful scenarios for the study and validation of new potential drugs for CF [33,57].

## Data Availability

The authors confirm that the data supporting the findings of this study are available within the article.

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
