# Peer review of "Lumacaftor and Matrine: Possible Therapeutic Combination to Counteract the Inflammatory Process in Cystic Fibrosis"

_biomolecules, 2021, doi:10.3390/biom11030422_

Round 1
Reviewer 1 Report
This is a remarkably well-written submission and the data is interesting However, I do have few comments. Major concern: 1) The authors demonstrated that Matrine rescued F508del-CFTR at the apical membrane, like VX-809. Are the author suggesting that Matrine is a CFTR corrector? 2) The authors are suggesting that the synergistic effect of VX-809 and Matrine could reduce the inflammatory process. However, the authors did not provide evidence for this. The authors would have include some data to support this claim (i.e. expression of IL-8 +/- Matrine in A549 cells), or reduce the emphasis on inflammation. Moreover, please include a sentence about the publication by Stanton (PMID: 26018799) where the authors demonstrated that VX-809 has no effect on IL-6 and IL-8 secretion. Minor concern: Page 1, lane 10: please remove Baile and co-workers. Usually authors are not included in the abstract. Page 2, lines 71-74: please include a paragraph talking about the recent FDA approved drug combination (Trikafta) (PMID: 30334692). Moreover, please include two recent publications demonstrating that Trikafta could rescue rare mutations and interestingly, elexacaftor (VX-445) is also a CFTR potentatior. (PMID: 32853178, PMID: 33303536). Page 2, line 93: please use “Na+/K+ ATPase” rather than “Na+/K+ pump”. Moreover, are you sure that the sc-58475 is correct and not ab-58475 or sc-48345? Page 3, line 103: please use “Adenocarcinomic human alveolar basal epithelial cells”. There are many different types of “lung cells”. Page 3, line 116: please use 104 rather than 104, same for lines 127, 140. Page 3, line 147: please specify the % of the acrylamide. Page 4, line 169: "A549 cells" rather than "Human lung cells" Figure 1A: “C” is the control? Is it DMSO? Please specify in the figure legend Figure 1 C: please use a better representative blot with the same amount of the loading control. There are no evidence that Matrine rescued band C. Moreover, is this blot a representative of n=1 or n=3? Please specify and include the densitometric analysis. Figure 1C: please use Na+/K+ ATPase Page 8, line 284: “These data likely indicate that the synergy between Vx-809 and Matrine…” Please check the stats between Vx and Vx+M to confirm the synergistic effect. Page 11, line 346: please discuss the publication by Bagdany et al (PMID: 28855508) about HSP90 and Hsc 70 Page 11: since the authors discussed infection/inflammation and neutrophil recruitment, is it important that they mention that infection by P. aeruginosa (native status of CF airway) reduced Orkambi-mediated F508del-CFTR rescue in airways epithelial cells (PMID: 26018799, PMID: 25792634, PMID: 32092967)?Author Response
Please see the attachment

Reviewer 2 Report
The authors follow up previous work on the effect of Matrine as modulators of HSC70 levels and rescue of the F508del protein (ref 24). The rationale of the work is of great interest to readers, but I fear that a thorough review is required before being accepted for publication.
Major revisions
Most of the refs in the introduction section should be updated, as much of the knowledge about cystic fibrosis has changed a lot since the discovery of the gene in 1989, see ref 5, ref 3 does not seem appropriate and ref 12 is no longer correct, as today we know that F508del mutation causes misfolding as well as gating protein problems.
Major revisions are to be made in the first session of the results “Vx-809 and Matrine co-treatment increases CFTR localization in the membrane” where:
- The authors stated that fluorescence of YFP reduces when the permeability of the CFTR channel is restored citing ref 30, but this happens when the cells expressing the CFTR protein, wild-type or mutated, are treated with forskolin or CTP-AMP, necessary to activate the CFTR channel, in figure 1A or in the text is not indicated such a treatment. Moreover, the CFTR localization should be demonstrated with some subcellular markers (for example RE, Golgi, and membrane markers).
- It is not clear what authors state with cytosolic CFTR fraction, and how flow cytometry can help distinguish cells with CFTR protein localized in the cytosol from those with CFTR located on the plasma membrane. In figure 1B, it is shown the percentage of CFTR positive cells, it is somewhat strange that only 20-30 percent of cells are positive for CFTR, speaking of a cell line stably expressing this protein.
- Figure 1C, the authors say that the treatment with Vx-809 and Matrine induces an increase of CFTR C band, as a mature form of the protein. This is correct, so, why is band B, representing the immature form, not present in the cytosolic extract fraction? Perhaps this type of experiment had to be done trying to quantize the fraction of CFTR localized on the plasma membrane, through biotinylation of membrane proteins and their immune-precipitation, also because, from the beginning of the synthesis to its localization on the plasma membrane, the CFTR protein is always localized on some type of membrane, endoplasmic reticulum, Golgi and so on.
Reviewer 3 Report
The manuscript can be accepted after major revisions. Please find my comments as attached file

Round 2
Reviewer 1 Report
The authors have done a great job in addressing all of my concerns with additional experiments and reworking their manuscript
Author Response
English language spelling and text editing was checked and the reference Eur. J. Med. Chem (2020), 204, 112631 was added as requested by one reviewer.
Reviewer 3 Report
The authors have improved the manuscript according to my suggestions.
Please include reference Eur. J. Med. Chem (2020), 204, 112631
as suggested in the previous revision process.
Author Response
English language spelling and text editing was checked and the reference Eur. J. Med. Chem (2020), 204, 112631 was added as requested by reviewer.